# Methodology to estimate natural- and vaccine-induced antibodies to SARS-CoV-2 in a large geographic region

Stacia M. DeSantis[1], Luis G. León-Novelo[1]*, Michael D. Swartz[1], Ashraf S. Yaseen[1], Melissa A. Valerio-Shewmaker[2], Yashar Talebi[1], Frances A. Brito[1], Jessica A. Ross[1], Harold W. Kohl III[3], Sarah E. Messiah[4,5], Steve H. Kelder[6], Leqing Wu[1], Shiming Zhang[1], Kimberly A. Aguillard[1], Michael O. Gonzalez[1], Onyinye S. Omega-Njemnob[6], David Lakey[7], Jennifer A. Shuford[8], Stephen Pont[6,8], Eric Boerwinkle[1]

1 The University of Texas Health Science Center at Houston, School of Public Health, Houston, Texas, United States of America, 2 The University of Texas Health Science Center at Houston, School of Public Health, Brownsville Campus, Brownsville, Texas, United States of America, 3 The University of Texas at Austin, Austin, Texas, United States of America, 4 The University of Texas Health Science Center at Houston, School of Public Health, Dallas Campus, Dallas, Texas, United States of America, 5 Center for Pediatric Population Health, UTHealth School of Public Health and Children's Health System of Texas, Dallas, Texas, United States of America, 6 The University of Texas Health Science Center at Houston, School of Public Health, Austin Campus, Austin, Texas, United States of America, 7 The University of Texas System and the University of Texas at Tyler Health Science Center, Tyler, Texas, United States of America, 8 Texas Department of State Health Services, Austin, Texas, United States of America

* Luis.G.LeonNovelo@uth.tmc.edu

**Data Availability Statement:** Summary information of these data can be found at https://sph.uth.edu/projects/texascares/dashboard

## Abstract

Accurate estimates of natural and/or vaccine-induced antibodies to SARS-CoV-2 are difficult to obtain. Although model-based estimates of seroprevalence have been proposed, they require inputting unknown parameters including viral reproduction number, longevity of immune response, and other dynamic factors. In contrast to a model-based approach, the current study presents a data-driven detailed statistical procedure for estimating total seroprevalence (defined as antibodies from natural infection or from full vaccination) in a region using prospectively collected serological data and state-level vaccination data. Specifically, we conducted a longitudinal statewide serological survey with 88,605 participants 5 years or older with 3 prospective blood draws beginning September 30, 2020. Along with state vaccination data, as of October 31, 2021, the estimated percentage of those 5 years or older with naturally occurring antibodies to SARS-CoV-2 in Texas is 35.0% (95% *CI* = (33.1%, 36.9%)). This is 3× higher than, state-confirmed COVID-19 cases (11.83%) for all ages. The percentage with naturally occurring or vaccine-induced antibodies (total seroprevalence) is 77.42%. This methodology is integral to pandemic preparedness as accurate estimates of seroprevalence can inform policy-making decisions relevant to SARS-CoV-2.

## Introduction

It is increasingly important to estimate the percentage of individuals in the US who have circulating anti-SARS-CoV-2 antibodies. People obtain protection through either natural infection

Data are available from the UTHEALTH Institutional Data Access / Ethics Committee (contact via email TexasCares@uth.tmc.edu) for researchers who meet the criteria for access to confidential data.

**Funding:** Project funded by Texas Department of State Health Services (Grant #HHS000866600001). Funding partners met/meet weekly with the state funding team to review survey findings and two representatives of the funding team are also co-authors on this manuscript.

**Competing interests:** The authors have declared that no competing interests exist.

with SARS-CoV-2 or full vaccination, and seroprevalence is the combination of these two avenues. Typically, an estimate of seproprevalence is obtained using mathematical modeling and simulation, which require inputs such as duration of immunity once infected, viral reproduction rate, population mixing, and additional factors [1–5]. However, the contributions of these inputs are still not fully known. For example, researchers are unsure of the duration of immunity afforded by natural infection or vaccination, and possible T-cell cross-reactivity. Further, continual emergence of SARS-CoV-2 variants make the picture even less clear [6–8].

Recent research suggests neutralizing and nucleocapsid antibodies to SARS-CoV-2 persist for at least 5–6 months [9–12] or possibly longer [13], and that re-infection risk is low in the several months after initial infection [14]. This resultant expected reduction in viral spread from these assumptions has inspired the idea of a path to normality. [15].

Given the above most current prevailing assumptions that: 1. Reinfection with COVID-19 within a few months is rare [16]; 2. Antibodies from natural infections typically last at least 5 months and cross-reactivity of serological tests is rare [11, 12, 17, 18]; and, 3. Vaccination produces a robust and reasonably long-term antibody response, making it possible to estimate regional seroprevalence as a combination of detection of natural antibodies, and state-level vaccination data [19].

The goal of this report is both to demonstrate this estimation process using a prospectively designed serological survey, as well as to provide the overall seroprevalence. To this end, we first estimate period seroprevalence over 1-week intervals from blood specimens collected prospectively from 88,605 community-based participants throughout Texas. We then compute a census age-adjusted seroprevalence estimate based on serum samples, and combine this with the Texas Department of State Health Services (DSHS) de-identified population-level vaccination counts https://www.arcgis.com/apps/dashboards/45e18cba105c478697c76acbbf86a6bc. Notably, the approach is not limited to the current pandemic; it is applicable to any infectious disease.

## Methods

### Participants and study design

The Texas Coronavirus Antibody REsponse Survey (Texas CARES) initiative has been previously described [20–22]. Briefly, Texas CARES is a prospective convenience sample of adult retail/business employees, K-12 and university educators and university students, those attending Health Resources and Services Administration (HRSA)- designated federally qualified health centers (FQHCs), and children 5–17 years, all of whom agreed to longitudinal monitoring of SARS-CoV-2 antibody status every three months (three time points) from 10/1/2020–10/31/2021. All protocols were reviewed and approved by the University of Texas Health Science Center's Committee for the Protection of Human Subjects, and were also deemed public health practice by the Texas Department of State Health Services IRB. All adults consented electronically to be in the study, and children under 18 either consented or assented to be in the study. The catchment area for the cohort study was the entire state of Texas. Recruitment efforts were taken to enroll rural and urban participants spanning the state. More details about the study are publicly available on the Texas CARES dashboard [23].

### Serological assay and state vaccination records

Antibody status was determined using the Roche Elecsys® Anti-SARS-CoV-2 (qualitative) assay detection of antibodies against SARS-CoV-2 nucleocapsid (N) protein, hereafter referred to as "Roche N-test". Based on Roche guidance, a positive result was assumed to be indication

that natural infection had occurred. The test has a sensitivity (95% confidence interval, CI) of 99.5% (97.0,100.0) and specificity of 99.82%(99.69, 99.91)≥14 days after infection.

De-identified population level daily vaccination data (2 doses for mRNA vaccines or 1 dose for Johnson and Johnson vaccine) by age group were obtained from Texas Department of State Health Services (DSHS). As of Oct 31, 2021 55.9% (8.65 million) Texans received at least 2 doses Pfizer, 36.4% (5.64 million) received at least 2 doses of Moderna, and 7.7% (1.19 million) received at least one dose of Johnson and Johnson. Given this breakdown, our results largely reflect mRNA-induced antibodies. https://data.cdc.gov/Vaccinations/COVID-19-Vaccinations-in-the-United-States-Jurisdi/unsk-b7fc/data, Accessed July 13, 2022.

## Statistical methods

We estimate the following components from the data, (1) presence of antibodies from natural infection, (2) presence of antibodies from complete vaccination (it is assumed full vaccination results in antibodies in all individuals), and (3) total antibodies (natural- or vaccine-induced antibodies). Natural antibody period seroprevalence is calculated at a given time interval of Texas CARES, while vaccine-induced antibodies are assumed known, as recorded by DSHS. A 1-week interval was deemed appropriate given the participant accrual rate into Texas CARES and disease wave fluctuations. Within this interval, we assume serological status from prior infection and vaccine status are not independent events. This assumption is very well-supported by the data, which as of October 31st indicate 16.52% of those with reported prior documented COVID-19 disease are vaccinated, versus 80.88% without prior documented COVID-19. Our data and other research [24, 25] support that presence of natural and vaccine-induced antibodies are very likely not independent events.

**Calculating total antibodies to SARS-CoV-2 in Texas.** Let $H$ denote the total number of census age groups, $h = 1, \ldots, H$. Assume we have a serological sample and true vaccination data for $T$ weeks. Let $t$ index the time window (week), where $t = 1, \ldots, T$. Now define:

- $v_{ht}$: Vaccination proportion in age group $h$ at week $t$ (provided by state records). Since the vaccination status is cumulative, $v_{h1} \leq v_{h2} \ldots \leq v_{hT}$ for $h = 1, \ldots, H$, and $v_{ht}$ is known with certainty.

- $\eta_{ht}$: Proportion with natural antibodies in age group $h$ at week $t$. This is unknown but is estimated cross-sectionally using the Roche N-test from Texas CARES.

- $w_h$: Proportion of the Texas population in age group $h$. Thus, $w_h$ is also known, and $\sum_h w_h = 1$.

The rate in time window $t$ (defined as having received the vaccine or testing positive for antibodies in the time window) in age group $h$ is given below, where "natural" refers to antibodies detected in the sample from natural infection and "vaccine" refers to antibodies produced by vaccination, fully known from state records,

$$
\begin{aligned}
\iota_{ht} &= P[\text{natural or vaccine}] \\
&= P[\text{natural }] + P[\text{vaccine } \& \text{ no natural}] \\
&= P[\text{natural }] + P[\text{no natural } | \text{ vaccine}]P[\text{vaccine }] \\
&= \eta_{ht} + \kappa_h v_{ht}.
\end{aligned}
\tag{1}
$$

In Eq (1) above, for brevity, we omit the text "in group age $h$ at week $t$". The definition of

$$
\kappa_h := P[\text{no natural antibodies in group } h \text{ at week } t \mid \text{vaccine}].
$$

This implicitly assumes the probability is equal across all weeks, $t = 1, \ldots, T$. Also notice that

the proportion of the population with both natural- and vaccine- induced antibodies is easily estimated as $(1 - \kappa_h)v_{ht}$, so a mathematically equivalent expression to Eq (1) is

$$
\begin{aligned}
\iota_{ht} &= P[\text{natural or vaccine}] \\
&= P[\text{natural}] + P[\text{vaccine}] - P[\text{natural \& vaccine}] \\
&= \eta_{ht} + v_{ht} - (1 - \kappa_h)v_{ht},
\end{aligned}
\tag{2}
$$

which may appear more intuitive: the total antibody rate is equal to the sum of the natural- and vaccine-induced antibody rates in a time interval, minus their overlap. Thus the estimated population seroprevalence at week $t$ is,

$$
SPR_t = \sum_h w_h \eta_{ht},
\tag{3}
$$

and the total rate (natural or vaccine induced) at week $t$ in the population is,

$$
IR_t = \sum_h w_h \iota_{ht} = \sum_h w_h (\eta_{ht} + \kappa_h v_{ht}),
\tag{4}
$$

where $\iota_{ht}$ is given in Eq (1). The population proportion with both natural and vaccine-induced antibodies is,

$$
\sum_h w_h (1 - \kappa_h) v_{ht}.
$$

In order to estimate $SPR_t$ and $IR_t$ we must estimate $\eta_{ht}$ and $\kappa_h$. We show these steps in the following subsubsection. We also note that had we assumed independence between natural- and vaccine-induced antibodies, $\kappa_h = 1 - \eta_{ht}$, as expected.

**Estimation of parameters for calculation of total antibodies.** We estimate $\kappa_h$ and $\eta_{ht}$ using the Roche N-test results. First, $\kappa_h$ is estimated using the information from all $T = 43$ study weeks since January 1, 2021, as the following sample proportion,

$$
\widetilde{\kappa}_h = \frac{\# \text{ of participants without natural antibodies \& vaccinated in age group } h \text{ at any week}}{\# \text{ of vaccinated participants in age group } h},
\tag{5}
$$

and $\eta_{ht}$ is initially estimated as,

$$
\dot{\eta}_{ht} = \frac{\# \text{ participants in group } h \text{ tested at week } t \text{ with natural antibodies}}{\# \text{ participants in group } h \text{ tested at week } t}.
\tag{6}
$$

Once we have $\dot{\eta}_{i1}, \cdots, \dot{\eta}_{iT}$ (and the denominator in the Eq (6) above) we compute the isotonic version (across index $t$) of $\dot{\eta}_{ht}$, $\widetilde{\eta}_{ht}$ such that $\widetilde{\eta}_{h1} \leq \widetilde{\eta}_{h2} \leq \cdots \leq \widetilde{\eta}_{hT}$ for $h = 1, \ldots, H$. See S.1. in S1 Appendix details of this calculation. The isotonic estimate of $\eta_{ht}$ is appropriate here because it reflects the fact that seroprevalence should not decrease over a short time interval (even though its raw estimate $\dot{\eta}_{ht}$ can decrease due to expected sampling error in a small window, $t$). Once these estimates are obtained, we compute the estimates of $SPR_t$ and $IR_t$, called $\widetilde{SPR}_t$ and $\widetilde{IR}_t$, by substituting the values of $\widetilde{\eta}_{ht}$ and $\widetilde{\kappa}_h$ into Eqs (3) and (4). Construction of a 95% confidence interval for $\widetilde{SPR}_t$ is based on that for a proportion from a stratified design in which the outcome variable is binary [26, 27] (details provided in S.2 in S1 Appendix).

**Algorithm to estimate the total antibody curve from January 1, 2021 to October 31, 2021.** Recalling, $H$ is the total number of age groups and $T$ the total number of weeks. The algorithm is:

1. For, $h = 1, \ldots, H$

a. Using the Roche N-test, compute $\widetilde{\kappa}_h$ in Eq (5).

b. Obtain the (cumulative) state vaccination rate for week $t$ by age group, denoted $v_{ht}$, from the Texas Dept of State Health Services database. Since they are cumulative, $v_{h1} \leq v_{h2} \leq \ldots \leq v_{hT}$ for $h = 1, \ldots, H$.

c. For $t = 1, \ldots, T$, compute the preliminary estimated N-test positive rate in the study at week $t$, $\dot{\eta}_{ht}$ in Eq (6). Next, compute the isotonic version of $\dot{\eta}_{ht}$, $\widetilde{\eta}_{ht}$, such that $\widetilde{\eta}_{h1} \leq \widetilde{\eta}_{h2} \leq \cdots \leq \widetilde{\eta}_{hT}$.

2. Estimate the age-adjusted seroprevalence rate at week $t$,

$$\widetilde{SPR}_t = \sum_h w_h \times \widetilde{\eta}_{ht},$$

and then the total seroprevalence at week $t$,

$$\widetilde{IR}_t = \sum_h w_h \times [\widetilde{\eta}_{ht} + \widetilde{\kappa}_h v_{ht}].$$

3. Plot $t$ v.s. $\widetilde{SPR}_t$ and $t$ v.s. $\widetilde{IR}_t$. These are Figs 1 and 2 discussed below.

## Results

Descriptive information for the full sample, and adults 18 years and over, respectively, are shown in Tables 1 and 2. Table 1 also includes seropositivity by demographics for the period Oct 1, 2021- Oct 31, 2021. The mean (standard deviation) age of all participants was 50.5 years (16.1) and most participants were in the 50–64 year age group (31.2%). Most were female (67.2%), White (90.4%), and from urban locations (93.2%). Most adults reported having some college education or an advanced or professional degree, and were employed full time. Table 1 shows differences in seropositivity by age and race, but not by sex or ethnicity. More granular demographic, sociodemographic, and spatial data are available on the TX CARES Dashboard (https://sph.uth.edu/projects/texascares/dashboard, accessed 7/12/22).

We applied the method with $H = 10$ age groups, partially informed by vaccine rollout ages, 5–15, 16–17, 18–29, 30–39, 40–49, 50–64, 65–74, 75–79, 80–84 and 85+ years. Note that $w_h$ is then the proportion of Texas population 5 years or older in the age group $h$. The census age-adjusted Texas COVID-19 seroprevalence using the Roche N-protein test over time (i.e., $t$ v.s $\widetilde{SPR}_t$) along with the 95% pointwise confidence band is shown in Fig 1. The vertical line on the graph delineates the time of first vaccine availability. Surges in seroprevalence correspond well to the known waves of SARS-CoV-2 in Texas [28].

We note that $\widetilde{\kappa}_h \approx 0.85$ in all age groups; thus, the proportion of study participants who reported having had both COVID-19, and being fully vaccinated were roughly $1 - \kappa_h \approx 15\%$. This indicates a violation of independence of natural infection and vaccination, which was expected. As these people must not be counted twice in a seroprevalence estimate, they are subtracted appropriately in each time period (week) per Eq (2).

The estimated age-adjusted total period seroprevalence in Texas, defined as either having had natural SARS-CoV-2 infection or being fully vaccinated (solid line) over time (i.e., $t$ v.s $\widetilde{IR}_t$) is shown in Fig 2. As of October 31, 2021, this is estimated to be 80% of the Texas population, with approximately 35.0% (95% $CI = (33.1\%, 36.9\%)$ resulting from natural infection. To our knowledge, this is the most robust and accurate non-model based estimate to date in the state of Texas.

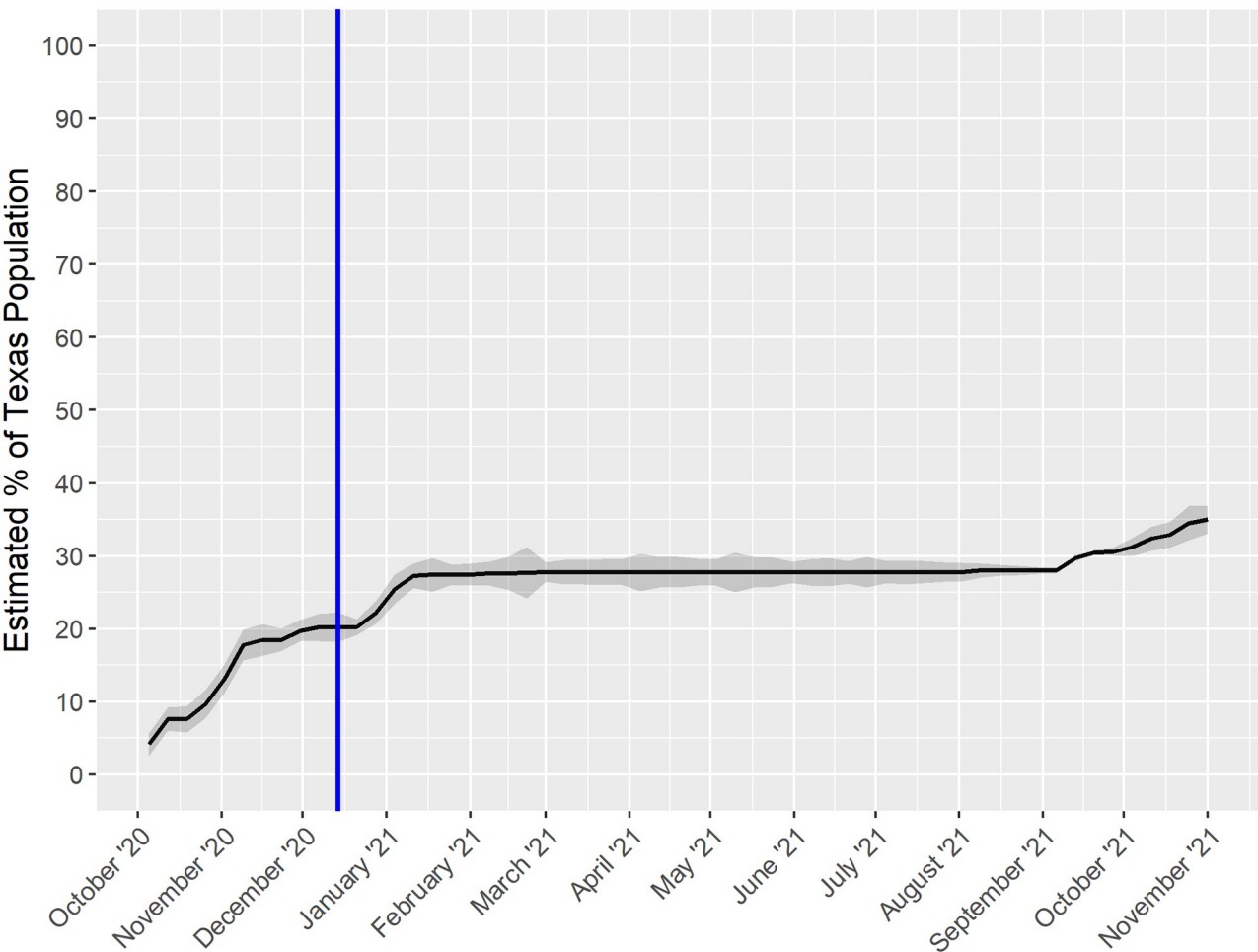

**Fig 1. Weekly SARS-CoV-2 natural antibody period seroprevalence and pointwise 95% confidence band for Texas CARES, estimated with isotonic transformed and with weighted seroprevalence age-standardized to the Texas 2021 census.** Blue vertical line indicates December 14, 2020, when vaccination started.

We do not include a confidence interval for total antibodies since the proportion vaccinated is a known (fixed) population quantity rather than an estimate, and thus does not lend itself to an estimate of variability. While the seroprevalence is not known or fixed, the large sample of 88,605participants would result in a very small range for the 95% confidence interval, if one were to be produced.

## Discussion

Using the methods proposed, the estimated proportion of the Texas population with antibodies against the SARS-CoV-2 virus, either from natural infection or induced by the vaccine, is nearly 80% as of October 31, 2021. This means 80% of the population benefits to some degree from reinfection from SARS-CoV-2 and acquiring COVID-19. There are several challenges to further practical or applied interpretation of these data. First, we do not fully understand the degree of protection from antibodies from natural infection compared to antibodies from the vaccines. The titer of antibodies from a full vaccine regimen is higher than a typical natural infection [29], but the diverse epitopes of a natural infection may offer advantages over antibodies targeting only spike protein. In addition, as SARS-CoV-2 mutates, new strains will

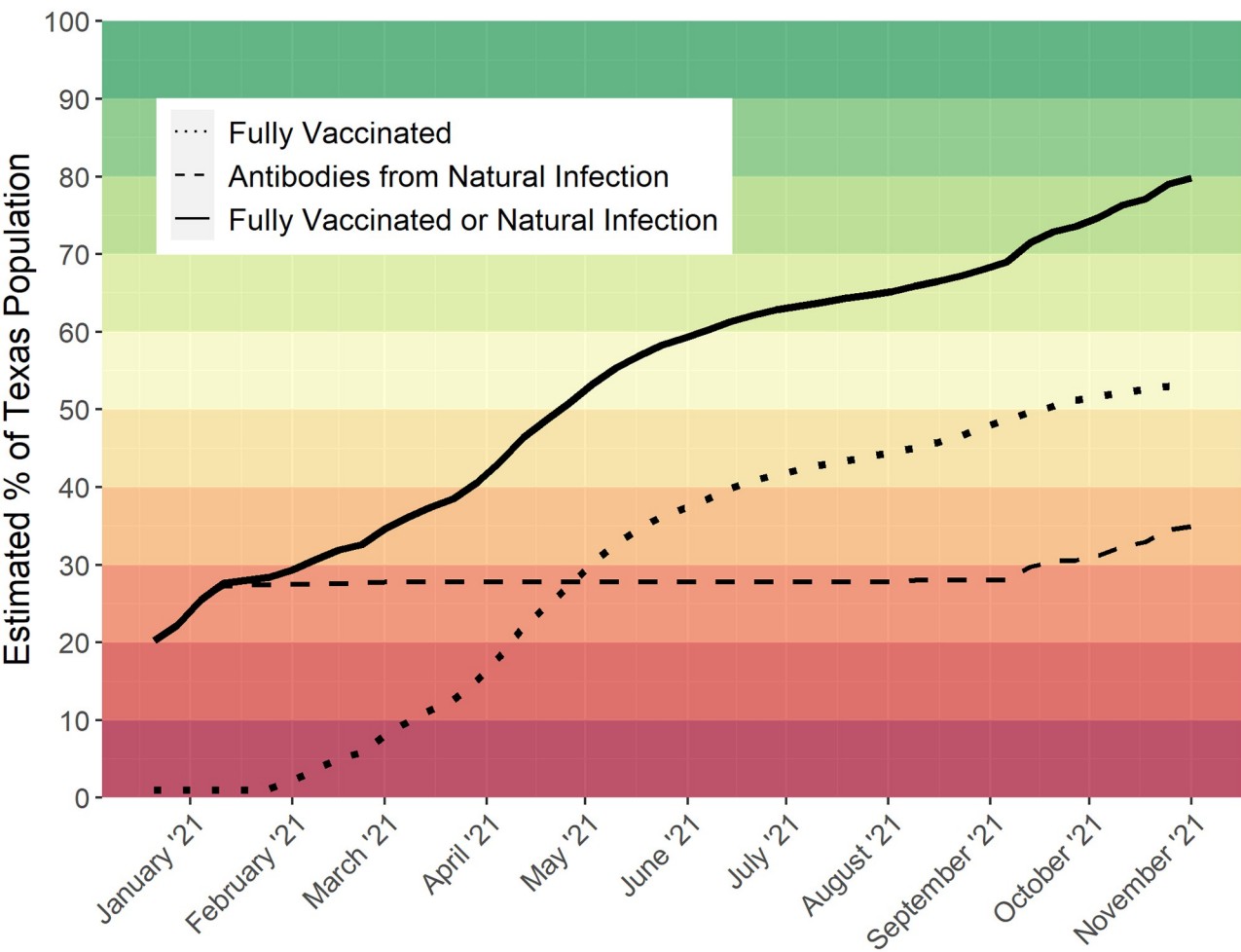

**Fig 2. Estimated natural- and vaccine-induced antibodies in Texas (i.e., weekly percentage of naturally occurring antibodies or fully vaccinated individuals).** The horizontal axis labels denote the first day of the month. The estimate as of October 31, 2021 is 77.42%.

likely influence the degree of protection of circulating antibodies [30]. Second, there is limited data on how long antibodies to the vaccine and to natural infection last, which makes it difficult to estimate total immunity [31–35]. Though early findings about the duration of natural and vaccine-generated antibodies are promising, it is reasonable to expect the proportion of people with detectable antibodies will decline over time. And finally, protection from an infectious agent is complex, and the concept of seroprevalence and protection does not take into account cell-mediated immunity and physical barriers, such as masks.

In contrast to model-based approaches, the current research will allow researchers and health departments to calculate regional estimate degree of likely immunity in the least biased manner. However, limitations occur in observational serological surveys; e.g., sample demographics may not be fully representative of the state, which is true for some variables in the current survey. Sampling variability or selection biases may operate overall, and/or within small time windows of a serological survey, and can result in inaccurate estimates of natural immunity in the region. It will therefore generally be necessary to smooth estimates using a chosen time window dependent on factors such as the magnitude of the wave of infection and participant accrual rate. Fortunately, we observe that the application of an isotonic restriction to reflect the assumption that seroprevalence should not decrease in a reasonably small time

**Table 1. Texas CARES participants' demographics overall, and seropositvity (period seroprevalence) from Oct 1, 2021 to Oct 31, 2021.**

| | Overall | Seropositivity | |
|---|---|---|---|
| | (N = 88605) | Negative 2318 (73.0%) | Positive 856 (27.0%) |
| **Age (Years)** | | | |
| Mean (SD) | 50.5 (16.1) | | |
| Missing | 0 | | |
| **Age (Years, categorized)** | | | |
| 5–15 | 1376 (1.6%) | 68 (51.5%) | 64 (48.5%) |
| 16–17 | 618 (0.7%) | 13 (52%) | 12 (48%) |
| 18–29 | 4805 (5.5%) | 106 (70.2%) | 45 (29.8%) |
| 30–39 | 14148 (16.2%) | 417 (75%) | 139 (25%) |
| 40–49 | 19032 (21.8%) | 486 (70.2%) | 206 (29.8%) |
| 50–64 | 27210 (31.2%) | 713 (72.2%) | 275 (27.8%) |
| 65–74 | 15760 (18.1%) | 432 (80.9%) | 102 (19.1%) |
| 75–79 | 3246 (3.7%) | 68 (89.5%) | 8 (10.5%) |
| 80–84 | 771 (0.9%) | 10 (76.9%) | 3 (23.1%) |
| 85+ | 186 (0.2%) | 5 (71.4%) | 2 (28.6%) |
| Missing | 1453 | 0 (0%) | 0 (0%) |
| **Gender** | | | |
| Female | 59494 (67.2%) | 1619 (72.8%) | 604 (27.2%) |
| Male | 29039 (32.8%) | 694 (73.5%) | 250 (26.5%) |
| None of these describe me | 15 (0.0%) | 0 (0%) | 0 (0%) |
| Missing | 57 | 5 (71.4%) | 2 (28.6%) |
| **Race** | | | |
| American Indian/Alaskan Native | 359 (0.4%) | 9 (90%) | 1 (10%) |
| Asian | 4574 (5.3%) | 146 (81.6%) | 33 (18.4%) |
| Black | 1899 (2.2%) | 37 (68.5%) | 17 (31.5%) |
| Hawaiian/Other Pacific Islander | 122 (0.1%) | 4 (80%) | 1 (20%) |
| Multi-racial | 1368 (1.6%) | 56 (78.9%) | 15 (21.1%) |
| White | 78123 (90.4%) | 2011 (72.5%) | 764 (27.5%) |
| Missing | 2160 | 55 (68.8%) | 25 (31.3%) |
| **Ethnicity** | | | |
| Hispanic | 12446 (14.6%) | 306 (70%) | 131 (30%) |
| Non-Hispanic | 73068 (85.4%) | 1943 (73.8%) | 689 (26.2%) |
| Missing | 3091 | 69 (65.7%) | 36 (34.3%) |
| **BMI (categorical)** | | | |
| Underweight | 1037 (1.2%) | 63 (78.8%) | 17 (21.3%) |
| Healthy | 30928 (36.2%) | 852 (75.9%) | 270 (24.1%) |
| Overweight | 28615 (33.5%) | 743 (72.1%) | 287 (27.9%) |
| Obesity | 24873 (29.1%) | 575 (69.6%) | 251 (30.4%) |
| Missing | 3152 (%) | 85 (73.3%) | 31 (26.7%) |
| **Geographic Location** | | | |
| Rural | 5755 (6.8%) | 63 (48.1%) | 68 (51.9%) |
| Urban | 78983 (93.2%) | 1418 (74.5%) | 486 (25.5%) |
| Missing | 3867 (%) | 837 (73.5%) | 302 (26.5%) |

**Table 2. Texas CARES participants' sociodemographics and employment for participants aged 18 and older.**

| Adults ≥18 years | Overall | (N = 85158) |
|---|---|---|
| **Education** | | |
| Some high school or less | 549 | (0.7%) |
| High school graduate/GED | 5356 | (6.5%) |
| Some college, no degree | 11729 | (14.2%) |
| Two or four year college level degree | 35511 | (42.9%) |
| Advanced professional or academic degree | 29711 | (35.9%) |
| Missing | 2302 | |
| **Employment Status** | | |
| Employed-full time | 46677 | (56.9%) |
| Employed-part time | 8180 | (10.0%) |
| Not currently employed/Unemployed | 14777 | (18.0%) |
| Other | 12349 | (15.1%) |
| Missing | 3175 | |
| **Employment Industry** | | |
| Accommodation and Food Services | 850 | (1.6%) |
| Administrative and Support and Waste Management | 409 | (0.8%) |
| Agriculture, Forestry, Fishing & Hunting | 358 | (0.7%) |
| Arts, Entertainment and Recreation | 1063 | (2.0%) |
| Central Administrative Office Activity | 1429 | (2.7%) |
| Construction | 1383 | (2.6%) |
| Educational Services | 7781 | (14.4%) |
| Finance and Insurance | 2848 | (5.3%) |
| Health Care and Social Assistance | 21034 | (39.0%) |
| Information | 1785 | (3.3%) |
| Management of Companies and Enterprises | 1305 | (2.4%) |
| Manufacturing | 1402 | (2.6%) |
| Mining | 158 | (0.3%) |
| Other | 61 | (0.1%) |
| Professional, Scientific and Technical Services | 7104 | (13.2%) |
| Real Estate and Rental and Leasing | 1345 | (2.5%) |
| Retail Trade | 1733 | (3.2%) |
| Transportation and Warehousing | 964 | (1.8%) |
| Utilities | 555 | (1.0%) |
| Wholesale Trade | 301 | (0.6%) |
| Missing | 31290 | |

window mostly overcomes the issue of daily or weekly sampling variability. Further, it is necessary to estimate the percentage of people who have both had natural COVID-19 infection and who are fully vaccinated in a given time window in order to subtract that proportion from the overall sum. It is also important to age-adjust estimated serological and vaccination rates to the state census so they are commensurate with population age structure. This is especially important since vaccination was rolled out by age group, with older adults first priority in January-March 2021 and approval for children and adolescents arriving much later in 2021. Finally, we could not adjust for all demographic factors, e.g., sex, race, ethnicity, to the distribution in the Texas population. This is due to the fact that, for example, for 10 age strata and 2 sex strata one would have 20 strata over 1-week periods of estimation, often resulting in no

participants in a given strata. Adding race and ethnicity adjustment, although ideal, would have further decreased strata size leading to unreliable statistical inference. While this is a limitation of the current study, it is notable that the natural antibody status was not different between men and women (26.5% and 27.2% respectively). However, there were race differences in antibody status, thus not standardizing remains a limitation of this study.

To our knowledge, this is the first fully data-driven estimation of infection- or vaccine-induced antibodies to SARS-CoV-2 in the state of Texas, which is the second largest state in the US with a population of 29.2 million. The estimated natural antibody rate of 35.0% contrasts with state-confirmed COVID-19 cases of 11.83%, demonstrating the importance of population-level studies. The method proposed and applied can be applied to any state or geographic area using vaccine counts, and an estimate of seroprevalence. As the pandemic unfolds and new variants are introduced, the estimates produced here will require further investigation.

## Supporting information

**S1 Appendix.**
(PDF)

## Author Contributions

**Conceptualization:** Stacia M. DeSantis, Luis G. León-Novelo, Michael D. Swartz, Ashraf S. Yaseen, Melissa A. Valerio-Shewmaker, Yashar Talebi, Harold W. Kohl, III, Sarah E. Messiah, Steve H. Kelder, Leqing Wu, Shiming Zhang, Kimberly A. Aguillard, Onyinye S. Omega-Njemnob, David Lakey, Jennifer A. Shuford, Stephen Pont, Eric Boerwinkle.

**Data curation:** Ashraf S. Yaseen, Frances A. Brito, Jessica A. Ross, Michael O. Gonzalez.

**Formal analysis:** Stacia M. DeSantis, Luis G. León-Novelo, Michael D. Swartz, Yashar Talebi, Frances A. Brito, Leqing Wu, Shiming Zhang.

**Funding acquisition:** Jennifer A. Shuford, Stephen Pont, Eric Boerwinkle.

**Investigation:** Onyinye S. Omega-Njemnob.

**Methodology:** Stacia M. DeSantis, Luis G. León-Novelo, Michael D. Swartz, Yashar Talebi.

**Project administration:** Ashraf S. Yaseen, Jessica A. Ross, Harold W. Kohl, III, Onyinye S. Omega-Njemnob, Stephen Pont, Eric Boerwinkle.

**Software:** Ashraf S. Yaseen, Yashar Talebi, Michael O. Gonzalez.

**Supervision:** Stacia M. DeSantis, Luis G. León-Novelo, Ashraf S. Yaseen, Melissa A. Valerio-Shewmaker, Jessica A. Ross, Harold W. Kohl, III, Sarah E. Messiah, Steve H. Kelder, Kimberly A. Aguillard, Michael O. Gonzalez, David Lakey, Jennifer A. Shuford, Stephen Pont.

**Validation:** Ashraf S. Yaseen, Michael O. Gonzalez, David Lakey.

**Visualization:** Ashraf S. Yaseen, Shiming Zhang, Michael O. Gonzalez, Eric Boerwinkle.

**Writing – original draft:** Stacia M. DeSantis, Luis G. León-Novelo, Michael D. Swartz, Ashraf S. Yaseen, Melissa A. Valerio-Shewmaker, Sarah E. Messiah, Steve H. Kelder, David Lakey, Jennifer A. Shuford, Stephen Pont, Eric Boerwinkle.

**Writing – review & editing:** Stacia M. DeSantis, Luis G. León-Novelo, Michael D. Swartz, Ashraf S. Yaseen, Melissa A. Valerio-Shewmaker, Yashar Talebi, Frances A. Brito, Jessica A. Ross, Harold W. Kohl, III, Sarah E. Messiah, Steve H. Kelder, Leqing Wu, Shiming Zhang,

Kimberly A. Aguillard, Onyinye S. Omega-Njemnob, David Lakey, Jennifer A. Shuford, Stephen Pont, Eric Boerwinkle.

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
