## [Decision Letter · Decision Letter 0]

23 Jun 2022

PONE-D-21-40593

Estimation of Natural- and Vaccine-Induced Antibodies to SARS-CoV-2 in a Large Geographic Region

PLOS ONE

Dear Dr. Leon Novelo,

Thank you for submitting your manuscript to PLOS ONE. After careful consideration, we feel that it has merit but does not fully meet PLOS ONE’s publication criteria as it currently stands. Therefore, we invite you to submit a revised version of the manuscript that addresses the points raised during the review process.

Please see detailed comments from the reviewers below. The expert reviewers request a number of clarifications regarding population sample/ demographics and request improvements to the reporting of methodological aspects of the study. Can you please carefully revise the manuscript to address all comments raised?

We look forward to receiving your revised manuscript.

Kind regards,

Katrien Janin

Staff Editor

PLOS ONE

a) Did participants provide their written or verbal informed consent to participate in this study?

“Project funded by Texas Department of State Health Services (Grant #HHS000866600001)”

5. We note that the grant information you provided in the ‘Funding Information’ and ‘Financial Disclosure’ sections do not match. When you resubmit, please ensure that you provide the correct grant numbers for the awards you received for your study in the ‘Funding Information’ section.

6. Thank you for stating the following in the Funding Section of your manuscript:

“Project funded by Texas Department of State Health Services (Grant #HHS000866600001)”

“Project funded by Texas Department of State Health Services (Grant #HHS000866600001)”

7. In your Data Availability statement, you have not specified where the minimal data set underlying the results described in your manuscript can be found. PLOS defines a study's minimal data set as the underlying data used to reach the conclusions drawn in the manuscript and any additional data required to replicate the reported study findings in their entirety. All PLOS journals require that the minimal data set be made fully available. For more information about our data policy, please see http://journals.plos.org/plosone/s/data-availability.

8. We note that you have indicated that data from this study are available upon request. PLOS only allows data to be available upon request if there are legal or ethical restrictions on sharing data publicly. For more information on unacceptable data access restrictions, please see http://journals.plos.org/plosone/s/data-availability#loc-unacceptable-data-access-restrictions.

Reviewers' comments:

Reviewer's Responses to Questions

**Comments to the Author**

1. Is the manuscript technically sound, and do the data support the conclusions?

Reviewer #1: Partly

Reviewer #2: Partly

2. Has the statistical analysis been performed appropriately and rigorously? 

Reviewer #1: I Don't Know

Reviewer #2: Yes

3. Have the authors made all data underlying the findings in their manuscript fully available?

Reviewer #1: No

Reviewer #2: No

4. Is the manuscript presented in an intelligible fashion and written in standard English?

Reviewer #1: Yes

Reviewer #2: Yes

5. Review Comments to the Author

Reviewer #1: This study describes a statistical method to combine serial anti-nucleoprotein antibody testing and vaccination data to estimate the prevalence of SARS-CoV-2 antibodies in Texas due to natural infection, vaccination and both. The estimates given are for the overall state, but it would be more valuable to have these at finer resolution, e.g. by age groups, gender, ethnicity, county level, etc if the data is available.

Methods:

State whether the Texas CARES population is from all over Texas or just confined to certain cities/counties.

Results

As mentioned in the discussion, the population sampled for antibody testing does not appear to be representative for the state as a whole (especially e.g. in gender [67% female] and ethnicity [90% white]). Would it therefore be useful to also adjust for these demographics in the calculations (which are just census-adjusted for age groups)?

Demographics other than age (e.g. ethnicity) may also be particularly important in vaccine uptake rates. Were demographic data available for the vaccinated population?

Can Fig 1 & Fig 2 be combined as they contain the same data?

This figure may also benefit from breaking down into demographic factors e.g. age and ethnicity to help identify target groups with high disease attack rates or high levels of susceptibility.

The abstract compared the estimated natural antibody rate of 35.2% with confirmed COVID-19 cases of 11.83%, but this was not mentioned in the paper itself. As the study included a large sample of 87,466, perhaps comparisons between seropositive and reported case reports may be possible for demographic-specific layers e.g. county, age, ethnicity, etc.

Reviewer #2: Dear Editor,

I have now read the manuscript entitled: “Estimation of Natural and Vaccine-Induced Antibodies to SARS-CoV-2 in a Large Geographic Region” (manuscript no: PONE-D-21-40593) by Desantis SM, Leon Novelo LG et al. The manuscript content exemplify the result of using one SARS-CoV-2 serological assay (the Roche Elecsys called the Roche N-test) described to show neutralizing antibodies against the SARS-CoV-2 nucleocapsid (N) protein. Serum samples from a large population of 87 466 individuals (age span 0 – 85+years) from the state of Texas, USA were tested (study participants presented in Tables 1 and 2).

The study results seem interesting, but a number of issues in the manuscript need to be clarified and better explained.

Comments and questions:

It is interesting that the used serological assay is aimed at showing neutralizing antibodies against SARS-CoV-2 virus, and still the viral antigen seem to be the nucleoprotein of the virus. The nucleoprotein is a internal protein of the SARS-CoV-2 virus, and not exposed sufficiently on the surface of the infectious virus particle, so it is difficult to understand how the detected antibodies can be virus-neutralizing?

Q1. The authors should explain in which way the serological responses in the blood/serum samples of participating individuals can be virus-neutralizing if the assay target protein was an internal nucleocapsid virus protein?

Q2a. The vaccinated individuals in the study were vaccinated with the S1-spike presenting vaccines. Vaccines used were either mRNA-vaccines or Johnson and Johnson vaccine. The authors should reveal which mRNA vaccines were used (Please provide manufacturer and proportion of study participants that were given which of the vaccines).

Q3. If the vaccine recipients in the study population were given S1-spike vaccine, and the used serological Roche-N-test contain only the nucleoprotein of SARS-CoV-2? How can the anti-S1-spike antibodies be made to react with the nucleoprotein if the nucleoprotein is not in the vaccine? Please clarify.

Q4. In Materials and Methods, paragraph 2.1 the study populations are presented as students, or professionals, or children 5-17 years. However, in Table 1 also children between age 0-4 is given. What was the reason for including these young children? Where their serum also tested in the serological assays?

Q5a. Since the authors present their study populations in Tables 1 and 2 it would have been valuable to see how large proportions of the different groups that contained naturally infected respectively vaccinated individuals?.

Q5b) Similarly, which proportion received only the Johnson and Johnson vaccine once, and which proportion/category of study participant that received the mRNA vaccine twice?

Figures:

Q6. In figure 1 (Weekly natural immunity) Texas CARES Roche N-test), in the table-text …. “Seroprevalence by Age group” is claimed to be shown. However, this information look quite unclear in the graph provided?? The authors should explain how readers should be able to identify the seroprevalence for all age-groups studied (this information would be very valuable to see and would significantly enhance the interest of the obtained data in this manuscript !!).

6. PLOS authors have the option to publish the peer review history of their article (what does this mean?). If published, this will include your full peer review and any attached files.

Reviewer #1: No

Reviewer #2: No

---

## [Author Response · Author response to Decision Letter 0]

3 Aug 2022

We attached a word file addressing the reviewers comments.

---

## [Editor Report · Decision Letter 1]

15 Aug 2022

Methodology to estimate natural- and vaccine-induced antibodies to SARS-CoV-2 in a large geographic region

PONE-D-21-40593R1

Dear Dr. Luis Novelo,

We’re pleased to inform you that your manuscript has been judged scientifically suitable for publication and will be formally accepted for publication once it meets all outstanding technical requirements.

Kind regards,

M. Kariuki Njenga

Academic Editor

PLOS ONE
---

## [Editor Report · Acceptance letter]

25 Aug 2022

PONE-D-21-40593R1 

Methodology to estimate natural- and vaccine-induced antibodies to SARS-CoV-2 in a large geographic region 

Dear Dr. Leon Novelo:

I'm pleased to inform you that your manuscript has been deemed suitable for publication in PLOS ONE. Congratulations! Your manuscript is now with our production department. 

Kind regards, 

on behalf of

Dr. M. Kariuki Njenga 

Academic Editor

PLOS ONE